# Hallucination Detection in LLMs: Fast and Memory-Efficient Fine-Tuned Models

Gabriel Y. Arteaga*[1,2], Thomas B. Schön[2], and Nicolas Pielawski[2]

[1]Department of Informatics, University of Oslo
[2]Department of IT, Uppsala University
gabrieya@uio.no, nicolas.pielawski@it.uu.se

## Abstract

Uncertainty estimation is a necessary component when implementing AI in high-risk settings, such as autonomous cars, medicine, or insurances. Large Language Models (LLMs) have seen a surge in popularity in recent years, but they are subject to hallucinations, which may cause serious harm in high-risk settings. Despite their success, LLMs are expensive to train and run: they need large amounts of compute and memory, preventing the use of ensembling methods in practice. In this work, we present a novel method that allows for fast and memory-friendly training of LLM ensembles. We show that the resulting ensembles can detect hallucinations and are a viable approach in practice as only one GPU is needed for training and inference. Code available at: https://github.com/Gabriel-Arteaga/LLM-Ensemble

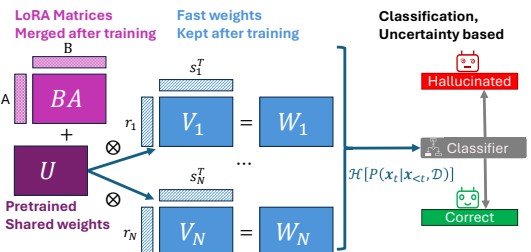

**Figure 1.** (Left) The ensemble utilizes a shared matrix of pre-trained "slow weights" $U$, which are updated with LoRA matrices ($BA$) during training and then merged. Each ensemble member is represented by an individual rank-one matrix (fast weights $V$) that is combined with the shared weights using a Hadamard product. (Right) The ensemble generates uncertainty estimates, which serve as features for a classifier to determine whether the LLM's prediction is correct or hallucinated.

## 1   Introduction

LLMs have recently grown in popularity, thanks to their ability to interpret natural language and generate answers that resemble human discussions, even surpassing human performance in specific tasks [1].

However, these models face a significant challenge known as *hallucination*, where outputs that seem plausible may either deviate from instructions or lack factual accuracy. Hallucinations can broadly be categorized into two types [2]: *faithfulness* hallucinations, where the LLM deviates from provided instructions, and *factual* hallucinations, where there is a disparity between the generated content and verifiable facts. The risk arises when individuals unaware of these limitations mistakenly treat such outputs as ground-truth, leading to decisions based on erroneous information — a concern particularly relevant to safety-critical areas such as healthcare.

Techniques leveraging natural language inference models and retrieval-based methods to detect hallucinations have shown promise in specific applications like summarization and open-domain question answering [3–5]. However, the effectiveness of these methods is typically limited to a narrow set of tasks, which restricts their generalizability across the broader spectrum of LLM applications.

Given these limitations, uncertainty estimation methods emerge as a compelling alternative for detecting both types of hallucinations [6]. Unlike task-specific approaches, uncertainty estimation uses the model's own confidence in its predictions to identify if the outputs are unfaithful or factually incorrect.

Recent work in uncertainty quantification in LLMs have emerged, with approaches like deep ensembles [7–10] and sample-based methods which use stochastic sampling techniques [11–15]. However, sample-based methods seldom provide reliable uncertainty estimates as they rely on the distribution of a single model's outputs, which may not fully capture the true uncertainty in the model's predictions. While deep ensembles advertise more robust uncertainty estimates by aggregating predictions from multiple independently trained models, they come with significant computational bottlenecks, especially when applied to larger LLMs, as they require substantial resources for training and inference.

To address these limitations, we propose a fast and memory-efficient deep ensemble method which is able to provide reliable uncertainty estimates. Figure 1 illustrates our proposed method, where Low-Rank Adaptation (LoRA) matrices are added on top of a pretrained model and used for fine-

---

*Corresponding Author.

Proceedings of the 6th Northern Lights Deep Learning Conference (NLDL), PMLR 265, 2025.

tuning. LoRA allows the whole ensemble to be trained cheaply and flexibly [16], with each ensemble member having an individual rank-one fast weight matrix[1]. This allows the members to be diverse. An uncertainty metric is used afterward and fed into a binary classifier, which is trained to discriminate between hallucinated and correct predictions.

We demonstrate that our method can effectively detect both factual and faithfulness hallucinations. Our method achieves a 97.8% accuracy in detecting faithfulness hallucinations, outperforming existing baselines. Additionally, our method attains a 68% accuracy in detecting factual hallucinations without compromising overall predictive performance. These results suggest that our method not only enhances the detection of hallucinations in LLMs, but also offers a practical solution for deploying these models in resource-constrained settings.

The main contributions of this paper are: **(1)** a fast and memory-efficient method for fine-tuning pre-trained LLMs using LoRA matrices and component-specific rank-1 matrix modifications, which reduces computational overhead and enables the effective use of ensemble methods; **(2)** a novel approach to hallucination detection that reformulates it as a binary classification task, leveraging uncertainty estimates from LLMs as features to distinguish between hallucinated and accurate content; and **(3)** demonstrating the practicality of our method for use with LLMs on minimal hardware, requiring only a single A40 GPU for both training and inference, thereby showcasing its efficiency and scalability.

## 2   Related Work

We believe that distinguishing between the two types of hallucinations introduced by Huang et al. [2] provides valuable insights into the behavior of LLMs and highlights the distinct challenges associated with each category. Therefore, we adopt this terminology throughout this paper.

**Hallucination Detection in LLMs**
Various approaches have been proposed to identify when an LLM diverges from instructions or deviates from contextual cues in the input. This work is especially critical in tasks like summarization, where adherence to the provided context is crucial for generating accurate summaries. These methods often leverage natural language inference models to compute entailment scores, which are then used to detect instances of unfaithfulness in the generated outputs [3, 5, 17].

Similarly, some research already focused on detecting when LLMs produce factual hallucinations,

where their generated content deviates from verifiable facts. Some methods have been developed for situations where the correct answer is known in advance, such as summarization and open-domain question answering [4, 18–21]. These approaches often involve comparing the generated content against a source document that is known to be accurate. When the ground-truth is not available, some methods leverage retrieval techniques to extract reliable information for verification [22, 23] or use LLMs themselves, using a prompt pipeline to facilitate hallucination detection [24].

Despite their effectiveness in specific contexts, many of these methods are constrained by their task-specific nature, limiting their generalizability across different LLM applications.

**Uncertainty Estimation Methods**
Uncertainty estimation methods are more general, offering greater versatility in hallucination detection [6]. Sample-based methods use sampling decoding techniques to introduce stochasticity in the LLM's responses, where a higher variance of the output is an indication of the model's uncertainty [13–15].

Another approach involves deep ensembles, which have been hypothesized to provide more informative uncertainty estimates compared to traditional methods [11, 13]. Deep ensembles leverage multiple model instances to capture a range of predictions, thus enhancing the robustness of uncertainty assessments [25]. However, implementing deep ensembles for LLMs through both pretraining [7, 8] and fine-tuning [9, 10] has primarily been constrained to smaller models. Scaling these ensembles to compete with state-of-the-art LLMs [26, 27] typically requires significant computational resources.

**Memory-Efficient Approaches**
To overcome the computational challenges associated with training deep ensembles of LLMs, recent research has focused on memory-efficient alternatives. One such approach, which serves as the backbone of our method, is BatchEnsemble [28], used to pre-train LLM ensembles more efficiently [29]. However, achieving state-of-the-art performance through pre-training can still be prohibitively expensive.

Recent studies have proposed a memory-friendly strategy that fine-tunes LLM ensembles from pre-trained weights, rather than training from scratch [30, 31]. This method, referred to as a LoRA Ensemble, assigns each ensemble member its own set of LoRA matrices [32]. While this approach has been utilized to compute uncertainty estimates [30, 31, 33], it has not been specifically applied to hallucination detection tasks. Our approach is similar in two ways: it reduces training costs by utilizing pre-trained weights, and it employs LoRA matrices during fine-tuning. However, unlike the LoRA Ensemble, our method does not rely on LoRA matrices to represent ensemble members. Instead, after

---

[1]LoRA matrices are rank-$r$ matrices added elementwise to the weight matrix, while the fast weights – rank-1 matrices – are multiplied elementwise.

fine-tuning, we merge the LoRA matrices with the pre-trained weights and represent the ensemble using sets of rank-one fast weights.

## 3 Method

**Uncertainty Estimation**
To quantify the uncertainty associated with an LLM's predictions, we use the predictive entropy of the output distribution, a concept rooted in information theory. Let $\mathbf{x}_{<t} = \{x_1, \ldots, x_{t-1}\}$ represent the preceding tokens, which serve as the input for predicting the target token $\mathbf{x}_t$ at time step $t$. The predictive entropy is then defined as:

$$\mathcal{H}\left[P(\mathbf{x}_t|\mathbf{x}_{<t}; \mathcal{D})\right] = \\ -\sum_{x_t} P(x_t|\mathbf{x}_{<t}; \mathcal{D}) \log P(x_t|\mathbf{x}_{<t}; \mathcal{D}) \quad (1)$$

where $\mathcal{D}$ refers to the overall training data distribution. The predictive entropy can further be divided into its two subcomponents aleatoric and epistemic uncertainties:

$$\underbrace{\mathcal{I}\left[\mathbf{x}_t, \theta|\mathbf{x}_{<t}, \mathcal{D}\right]}_{\text{Epistemic}} = \\ \underbrace{\mathcal{H}\left[P(\mathbf{x}_t|\mathbf{x}_{<t}; \mathcal{D})\right]}_{\text{Predictive}} - \underbrace{\mathbb{E}_{p(\theta|\mathcal{D})}\left[\mathcal{H}\left[P(\mathbf{x}_t|\mathbf{x}_{<t}; \mathcal{D})\right]\right]}_{\text{Aleatoric}}. \quad (2)$$

Epistemic uncertainty captures the lack of knowledge of a system, which shrinks as more data is made available. Conversely, aleatoric uncertainty represents the noise – the variability of the data – and is therefore irreducible [34].

$P(\mathbf{x}_t|\mathbf{x}_{<t}; \mathcal{D})$ cannot be directly computed and is approximated with:

$$P(\mathbf{x}_t|\mathbf{x}_{<t}; \mathcal{D}) \approx \frac{1}{M} \sum_{m=1}^{M} P(\mathbf{x}_t|\mathbf{x}_{<t}; \theta_m), \quad (3)$$

where $\theta_m$ is the parameters associated to the $m^{\text{th}}$ model. The estimate of the predictive entropy is an estimate of the predicted uncertainty, and will yield the exact predictive uncertainty as $M \to \infty$ [35].

**Memory-Efficient Fine-tuning**
We employ deep ensembles [25] to approximate Equation (3). However, training those deep ensembles may require prohibitively high computational resources, available to few organizations.

We propose using the BatchEnsemble method [28], but rather than pre-training each ensemble member [29], we adapt the method to fine-tune already pre-trained models. BatchEnsemble optimizes memory usage, which is a critical advantage over traditional ensembles. In a standard ensemble, memory requirements grow linearly with the number of ensemble members, with complexity increasing to $\mathcal{O}(Mmn)$ per layer, where $M$ is the number of

ensemble members and $mn$ represents the size of the weight matrices. In contrast, BatchEnsemble reduces memory complexity to $\mathcal{O}(mn + M(m + n))$ per layer, significantly lowering the memory footprint by sharing a single weight matrix $U$ across all ensemble members and augmenting it with trainable vectors $r_i \in \mathbb{R}^{m \times 1}$ and $s_i \in \mathbb{R}^{n \times 1}$. The outer product of these vectors yields a fast weight matrix $V_i$, allowing each ensemble member's weight matrix to be represented as the Hadamard product of the shared weight $U$ and the fast weight $V_i$:

$$W_i = U \odot V_i, \text{where } V_i = r_i s_i^T. \quad (4)$$

We further adapt this method by substituting the shared weight with a pre-trained weight, setting $U = \omega_{\text{pretrained}}$. To introduce diversity into the ensemble, we randomly initialize the fast weights. This initialization must be done carefully to preserve the knowledge stored in the pre-trained weights; initializing the fast weights with a mean of 1 is crucial to avoid disrupting the pre-trained knowledge. For more details on the weight initialization procedure, please refer to the Appendix.

To minimize the computational demands during fine-tuning, we apply the LoRA method [32]. We retain the pre-trained weight matrix $U$ as $U_0$ and introduce low-rank matrices $B \in \mathbb{R}^{m \times r}$ and $A \in \mathbb{R}^{r \times n}$, where $r \ll \min(m, n)$. This approach allows us to update $U$ as $U_0 + BA$, reducing the number of parameters that need to be trained while maintaining model performance. In contrast to the original LoRA paper [32], our method applies LoRA to all modules, which we believe enhances ensemble diversity and leads to more reliable uncertainty estimates.

**Noise Injection**
We hypothesize that injecting noise into the ensemble during training may enhance model diversity and improve uncertainty estimates. To explore this, we employ anchored ensembling [36] as one of our methods. However, this approach can be unstable when scaled to larger models. To address this issue, we incorporate a normalization term, as suggested in [6], to stabilize the training process.

**Hallucination Detection**
To detect hallucinations in an LLM, we design a subtask for a dedicated classifier. This task is framed as a binary classification problem, where the classifier is trained on a dataset containing uncertainty estimates from our ensemble and corresponding binary labels indicating whether the LLM is hallucinating or not. The choice of classifier depends on the practitioner's needs. For instance, one might choose a fast inference model like a shallow decision tree to minimize computational overhead, or opt for a more expressive model, such as a random forest, for enhanced performance.

# 4 Design of experiments

We aim to design experiments that can clearly differentiate between faithfulness and factual hallucinations, enabling us to assess the performance of our method for each type.

**Models and Baselines**
We evaluate our proposed adaptations to BatchEnsemble during fine-tuning, including BatchEnsemble with noise injection (NI). For uncertainty-based experiments, we compare against two baselines: a sample-based method that approximates the output distribution using repeated prompting with stochastic sampling (temperature = 0.5, top-p = 0.99, top-k = 5) [13, 15], and a LoRA Ensemble, which applies regularization to LoRA B matrices as described by Wang et al. on MMLU and SQuAD datasets [30].

All models, including baselines, are fine-tuned on all modules using LoRA adapters with a rank of $r = 8$ and a scaling factor of $\alpha = 32$. For BatchEnsemble, the LoRA matrices are merged into the shared weights after training. Unless otherwise specified, we use an ensemble size of 4 across all experiments. This ensemble size determines the number of rank-1 matrices for BatchEnsemble, the number of adapters for the LoRA ensemble, and the number of samples generated by the sample-based method. Additionally, all models, including baselines, use the uncertainty measures described in Equation (2).

We also evaluate our methods against a single model fine-tuned from pre-trained weights to ensure fair performance comparison. All models in this study leverage Mistral-7B-Instruct-v0.2 as a core component, either by directly fine-tuning the pre-trained weights or by incorporating them into more complex architectures like BatchEnsemble, where the shared weights are replaced by the pre-trained weights [26].

**Faithfulness Hallucination Detection**
To detect faithfulness hallucinations, we use the SQuAD and SQuAD 2.0 datasets [37, 38]. The datasets consist of contexts and questions, with SQuAD featuring answerable questions and SQuAD 2.0 including unanswerable ones. We instructed the LLMs to respond with "I don't know" if the answer was not in the context. Any other response to unanswerable questions indicated a faithfulness hallucination. Initially, we trained the models only on answerable questions, but this led to hallucinations across all test points. We then adjusted our approach by including 1/3 unanswerable questions in training to balance the model's responses.

**Factual Hallucination Detection**
For factual hallucination detection, we use the MMLU dataset [39], as the pre-trained Mistral 7B model [26] used it as a benchmark without training on it. MMLU contains multiple-choice questions from diverse knowledge areas. Models were instructed to select one of the available choices; incorrect answers were labeled as factual hallucinations.

**Predictive Performance**
We assess the predictive performance of our models on downstream tasks to evaluate if improved uncertainty estimates impact model accuracy. Models are fine-tuned on SQuAD and MMLU datasets, and evaluated using the F1 score, exact match for SQuAD, and accuracy for MMLU [37, 39].

**Out-Of-Distribution Test**
To test out-of-distribution detection, all models are fine-tuned on answerable questions from SQuAD 2.0 and evaluated on unanswerable ones, assessing the models' capacity to recognize shifts in data distribution.

**Classifier Training Details**
To evaluate the quality of the uncertainty estimates produced by our method and the baseline approaches, we trained five distinct classifiers: k-Nearest Neighbors, logistic regression, decision tree, random forest, and support vector machine. These classifiers were trained to distinguish between hallucinated and non-hallucinated predictions using uncertainty estimates derived from the outputs of our method and the baselines.

For each task—faithfulness hallucination detection, factual hallucination detection, and Out-of-Distribution (OOD) detection—we created custom datasets based on the outputs of our method and the baseline approaches. In all cases, 5000 data points were propagated through the models to generate predictions, which were then annotated as hallucinated or non-hallucinated depending on the task-specific criteria. The resulting datasets were split into 80% for training and 20% for testing, with the split performed randomly using a seed of 42 to ensure consistency across experiments.

The annotation methods for hallucinations were task-specific. For faithfulness hallucination detection, as described earlier, any response to unanswerable questions in the SQuAD v2 dataset that deviated from "I don't know" was labeled as hallucinated. For factual hallucination detection, we used multiple-choice questions from the MMLU dataset, where incorrect answers were labeled as hallucinated and correct answers as non-hallucinated. For OOD detection, we followed the same annotation procedure as faithfulness hallucination detection, using the same set of unanswerable questions from SQuAD v2, with the primary difference being how our method and the baselines were trained for this task.

**Table 1.** Top-1 Accuracy from classifiers on faithful and factual hallucination detection and OOD test.

| Method | Faithfulness | Factual | OOD |
|---|---|---|---|
| (Ours) BatchEnsemble | **97.8** | 68.0 | 62.4 |
| (Ours) BatchEnsemble+NI | 96.5 | 66.9 | 61.9 |
| LoRA Ensemble | 92.5 | **73.9** | **63.3** |
| Sample-Based | 92.1 | 69.6 | 62.2 |

**Table 2.** Performance metrics on SQuAD and MMLU datasets. (NF=not fine-tuned)

| Dataset | SQuAD | | MMLU |
|---|---|---|---|
| Metric | Exact Match | F1 Score | Accuracy |
| NF Single Model | 7.7 | 37.2 | 0.0 |
| NF BatchEnsemble | 8.1 | 37.9 | 0.0 |
| Single Model | 85.1 | 92.1 | 56.3 |
| (Ours) BatchEnsemble | **85.9** | **93.4** | **56.7** |
| (Ours) BatchEnsemble+NI | 85.4 | 92.6 | 53.2 |
| LoRA Ensemble | 68.4 | 84.4 | 44.6 |

# 5 Results

**Hallucination and OOD Detection**

In Table 1, we report the top-1 classification accuracy achieved by the best-performing classifier on the uncertainty estimates from our baselines and models. Detailed classification accuracies for each classifier are provided in Appendix B.

All methods, including the baselines, demonstrate a relatively high accuracy—exceeding 92%—in detecting cases where the model faithfully hallucinates. Table 3 describes an example of our method's response to both answerable and unanswerable questions. It shows a notable increase in uncertainty when the model encounters an unanswerable question. The uncertainty sharply decreases after the model generates the first token, suggesting that once it commits to a hallucinated token, it becomes more prone to continue hallucinating—a phenomenon referred to as the "snowballing effect" [40]. More, the high accuracy achieved using the models' uncertainty estimates supports the idea that LLMs' internal states possess an inherent understanding of the generated, hallucinated content, a finding similar to that observed by Azaria & Mitchell [41].

For detecting factual hallucinations, the LoRA Ensemble's uncertainty estimates result in the highest accuracy. A possible explanation is that the high weight decay strategy applied to the LoRA $B$ matrix [30] introduces substantial stochasticity among the ensemble members, leading to improved uncertainty estimates. However, this enhanced expressiveness in uncertainty estimation comes at the cost of predictive performance, which will be discussed further in the next subsection. Additionally, while the sample-based method demonstrates slightly better performance than our proposed approach, this marginal improvement may be attributed to randomness during training or to the task itself. Producing uncertainty estimates for a single token (the choice in a multiple-choice setting) may inherently be easier for the sample-based method.

All methods generally performs worse when encountering OOD data points. This indicates that, while LLMs are versatile, uncertainty-based approaches still remain limited in their ability to detect hallucinations for examples that does not lie in-distribution, highlighting the need for further research on detecting hallucinations in OOD scenarios.

**Predictive performance**

Table 2 presents the predictive performance of all evaluated models. The results indicate that while all models require fine-tuning to achieve optimal performance on downstream tasks, our proposed BatchEnsemble method with LoRA fine-tuning on shared weights consistently outperforms each baseline across all metrics. Notably, the LoRA ensemble performs worse than the single model despite being an ensemble. This performance discrepancy is likely attributed to the significant weight decay strategy implemented by Wang et al. [30], which involved fine-tuning only the query and key modules. The combination of high regularization and fine-tuning all modules appears to result in suboptimal performance. Additionally, the results from BatchEnsemble+NI further suggest that implementing regularization strategies effectively in practice poses substantial challenges.

**Time and Memory footprint**

All experiments were conducted on a single A40 GPU. Figure 2 illustrates the performance of BatchEnsemble and highlights its advantages. On the left side of the figure, it is shown that as the number of ensemble members increases, the rate at which inference speed improves for BatchEnsemble is lower compared to the sample-based approach. This suggests that BatchEnsemble, by processing all ensemble members' predictions simultaneously in a single forward pass, is faster in inference than the baseline method. On the right side of the figure, the limitations of using a Vanilla ensemble [25] for uncertainty estimation in LLMs are demonstrated. Specifically, as the number of ensemble members increases, the parameter size grows linearly with the Vanilla method, whereas the increase in BatchEnsemble's parameter size remains negligible.

# 6 Discussion

This paper focused on exploring uncertainty-based methods for detecting hallucinations in LLMs, and demonstrating the possibility of faithfulness and factual hallucination detection. While comparing against methods beyond uncertainty estimation could provide valuable insights, no standardized benchmark currently exists for systematic comparison between uncertainty-based and other detection

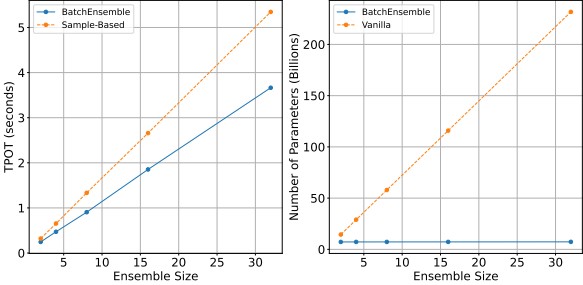

**Figure 2.** (left) The average time for the models' to output a token, as the ensemble size increases the BatchEnsemble becomes increasingly faster in inference compared to the baseline. (right) Trainable parameters increase linearly with ensemble size for Vanilla ensemble [25], while BatchEnsemble [28] shows negligible increase.

**(a)** Answer **is** available in context
Hallucination is **not** occurring

| tokens | pattern | recognition | re | cept | ors |
|---|---|---|---|---|---|
| **entropy** | 0.18 | 0.00 | 0.01 | 0.00 | 0.00 |

**(b)** Answer **is not** available in context
Hallucination is occurring

| tokens | P | ark | Sk | ary | sz | ew | ski |
|---|---|---|---|---|---|---|---|
| **entropy** | 2.02 | 0.03 | 0.56 | 0.0 | 0.0 | 0.0 | 0.0 |

**Table 3.** Per-token uncertainty (predictive entropy in bits) computed with the batch ensemble. (a) The context of the question contains all the information for providing a response. (b) The context does not contain the information needed for answering the question, the LLM hallucinates, instead of replying "I don't know".

strategies. We framed hallucination detection as a binary classification task and designed benchmarks accordingly. However, creating a comprehensive benchmark that incorporates other methods meaningfully remains a challenge. Addressing this gap presents a promising direction for future research.

Another area for future exploration is the use of alternative metrics beyond predictive entropy and its subcomponents. Methods such as EigenScore [15] or semantic entropy [13] may offer better performance for specific tasks by capturing uncertainty in more nuanced ways. Integrating these metrics into approaches like BatchEnsemble could enhance both efficiency and robustness of hallucination detection by reducing inference time while maintaining or improving reliability.

Our experiments were limited to exploring the method with a single pretrained language model. However, we hypothesize that the method works with other LLM sizes and architectures. Future work could explore the impact of factors such as model size on overconfidence and whether this is reflected in the model's uncertainty estimates, providing fur-

ther insights into the robustness and scalability of uncertainty-based hallucination detection.

Finally, we believe that aleatoric and epistemic uncertainty may correspond to faithfulness and factual hallucinations, respectively. Our findings suggest that current methods for measuring epistemic uncertainty are not diverse enough to confirm this hypothesis. Future work could involve developing more sophisticated noise injection techniques to generate diverse ensemble members, which may enhance the accuracy of epistemic uncertainty measurements while maintaining predictive performance. Such efforts could open new research directions to better understand how different uncertainty components relate to faithfulness and factual hallucinations.

# 7   Conclusion

We have developed a memory-efficient method to fine-tune LLMS using LoRA matrices and rank-1 matrix modifications. A key contribution of our work is demonstrating how pretrained models can be enhanced with an uncertainty estimation component to enable effective hallucination detection. Our approach demonstrated the ability to distinguish between hallucinated and non-hallucinated content, achieving high accuracy in faithfulness detection, even in resource-constrained settings.

While our uncertainty-based methods provide valuable tools for hallucination detection, and have shown strong performance in detecting faithfulness hallucinations, challenges remain, particularly in detecting factual hallucinations and managing out-of-distribution data points. Nevertheless, our approach is an important step towards reducing harmful outputs from LLMs, which is crucial for deploying these models in safety-critical environments.

The combination of uncertainty estimation with flexible fine-tuning approaches like BatchEnsemble and LoRA shows strong potential for enhancing LLM reliability and robustness. Future work could build on these findings by integrating more sophisticated uncertainty metrics and establishing standardized benchmarks, paving the way for more effective and generalizable hallucination detection strategies.

# Acknowledgments

This work builds upon research originally conducted as part of Gabriel Y. Arteaga's master thesis at Uppsala University [6].

This research was partially supported by the *Wallenberg AI, Autonomous Systems and Software Program (WASP)* funded by Knut and Alice Wallenberg Foundation, by *Kjell och Märta Beijer Foundation*, and ERC grant 101054643. The computations were enabled by resources provided by the National Aca-

demic Infrastructure for Supercomputing in Sweden (NAISS), partially funded by the Swedish Research Council through grant agreement no. 2022-06725.

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

# A    Experimental Details

In this section, we will outline the detailed training and evaluation splits used for our experiments

## A.1    Factual Hallucination Detection Experiment

For the factual hallucination detection experiment, we used the MMLU dataset  [39]. We created a training set by combining data from the *'all'* category and the *'auxiliary_train'* split. The dataset was shuffled with a seed of 42, and the first 42,000 data points were selected. Of these, 40,000 were used for training, while the remaining 2,000 were set aside for validation.

For evaluation, we used the *'test'* split, shuffled with the same seed of 42. We selected the first 5,000 data points from this split for evaluation.

## A.2    Faithfulness Hallucination Detection Experiment

For the faithfulness hallucination detection experiment, we used a modified version of the SQuAD v2 dataset. To create the training and validation splits, we first loaded the SQuAD v2 training set and filtered it into two subsets: answerable and unanswerable questions. Unanswerable questions were relabeled with "I don't know" to train the model to differentiate between the answerable and unanswerable questions.

We then shuffled both subsets with a seed of 50 and selected 28,000 answerable and 14,000 unanswerable questions to maintain a distribution similar to the original dataset. These subsets were combined into a single dataset and shuffled again to ensure a balanced distribution of answerable and unanswerable questions. From this combined dataset, we selected 40,000 samples for training and 2,000 samples for validation.

For evaluation, we processed the SQuAD v2 validation set and filtered to only keep unanswerable questions. The filtered unanswerable questions were shuffled with a seed of 42, and the first 5,000 data points were selected for evaluation.

## A.3    Out-Of-Distribution Detection Experiment

For the OOD detection experiment, we trained our models on the original SQuAD dataset and evaluated them on unanswerable questions from the SQuAD v2 validation set.

We began by loading the SQuAD training dataset and shuffling it using a seed of 50 to ensure randomness. From this dataset, we selected 40,000 samples for training and 2,000 samples for validation.

For evaluation, we utilized the validation set from SQuAD v2, specifically filtering out unanswerable questions. The unanswerable subset was shuffled with a seed of 42, and the first 5,000 data points were selected for the evaluation phase.

## A.4    Predictive Performance Experiments

To evaluate predictive performance, we used models trained during the previous experiments on both the MMLU and SQuAD datasets.

For the MMLU evaluation, we leveraged the models trained from the factual hallucination detection experiment. We loaded all data points from the MMLU test split, shuffled them with a fixed seed of 42, and selected the first 5,000 samples.

For the SQuAD dataset, we used models trained from the OOD detection experiment, as these were specifically trained on the SQuAD dataset. For evaluation, we loaded the SQuAD validation set, shuffled it with a seed of 42, and selected 5,000 samples.

# B    Detailed Results

The tables below present the detailed results for each method across five different classifiers: Logistic Regression (LR), Decision Tree (DT) classifier, Support Vector Classifier (SVC), Random Forest (RF), and k-Nearest Neighbors (kNN). All classifiers were implemented using the scikit-learn library [42]. No hyperparameter tuning was performed; instead, we used the default parameters provided by scikit-learn for each classifier.

For the features, we used the first token's predictive entropy and aleatoric uncertainty. Additionally, we provided the classifiers with two more features: the average predictive entropy and the average aleatoric

uncertainty. We also experimented with including epistemic uncertainty as a feature, but this reduced the performance, likely due to the correlation between the features. Therefore, we chose to exclude epistemic uncertainty and use the aforementioned features instead.

**Table B.1.** Classifier accuracies on faithfulness hallucination detection.

| Method | LR | DT | SVC | RF | kNN |
|---|---|---|---|---|---|
| BatchEnsemble | 89.7 | 96.8 | 92.7 | 97.8 | 95.9 |
| BatchEnsemble+NI | 85.0 | 95.8 | 91.0 | 96.5 | 96.5 |
| LoRA Ensemble | 89.9 | 89.2 | 90.4 | 92.5 | 92.1 |
| Sample-Based | 82.2 | 91.9 | 87.2 | 92.1 | 90.3 |

**Table B.2.** Classifier accuracies on factual hallucination detection.

| Method | LR | DT | SVC | RF | kNN |
|---|---|---|---|---|---|
| BatchEnsemble+NI | 66.31 | 58.99 | 66.92 | 60.06 | 61.59 |
| BatchEnsemble | 68.00 | 63.60 | 67.33 | 63.87 | 64.13 |
| LoRA Ensemble | 71.07 | 70.80 | 72.27 | 73.87 | 72.93 |
| Sample-Based | 69.60 | 61.33 | 69.60 | 61.33 | 65.33 |

**Table B.3.** Classifier accuracies on OOD detection.

| Method | LR | DT | SVC | RF | kNN |
|---|---|---|---|---|---|
| BatchEnsemble+NI | 61.9 | 56.95 | 61.8 | 60.75 | 59.15 |
| BatchEnsemble | 62.4 | 56.25 | 62.35 | 59.5 | 59.9 |
| LoRA Ensemble | 63.3 | 54.85 | 63.2 | 58.6 | 59.4 |
| Sample-Based | 62.15 | 54.5 | 61.6 | 58.05 | 59.45 |

# C   Weight Initialization

We explore two widely-used weight initialization strategies for our fast weights: He initialization [43] and Xavier initialization [44]. We hypothesize that the pre-trained Mistral 7B model [26] may have used one of these methods, making them natural choices for our approach.

The right side of Figure C.1 shows the generated responses for a sample data point from the SQuAD dataset, where the answer is nonsensical. This outcome is expected because the ensemble members are formed by the multiplicative product of shared and fast weights. Initializing the fast weights close to 0 effectively nullifies the knowledge embedded in the pre-trained weights. To address this, we adopt a simple solution: we maintain the variance of the He and Xavier initializations but set the mean to 1. This adjustment ensures that the ensemble members do not completely overwhelm the pre-trained weights.

The left side of Figure C.1 illustrates this modified approach. We observe that using He initialization with a mean of 1 produces diverse yet coherent predictions. The increased variability results from the higher variance of He initialization compared to Xavier initialization. We argue that greater diversity in the ensemble enhances uncertainty estimates. Therefore, we select He initialization with a mean of 1, as shown in option (a) of Figure C.1.

# D   Examples of prompts and answers

For the SQuAD and SQuAD v2 datasets we format our prompts to the LLMs using the following template:

**Table B.4.** Negative Log-Likelihood (NLL) results for various models on the SQuAD testset. (NF=not fine-tuned)

| Model | NLL |
|---|---|
| **NF Single Model** | 2.78 |
| **NF BatchEnsemble** | 2.77 |
| **Single Model** | **1.39** |
| **BatchEnsemble** | **1.39** |
| **BatchEnsemble+NI** | 1.40 |

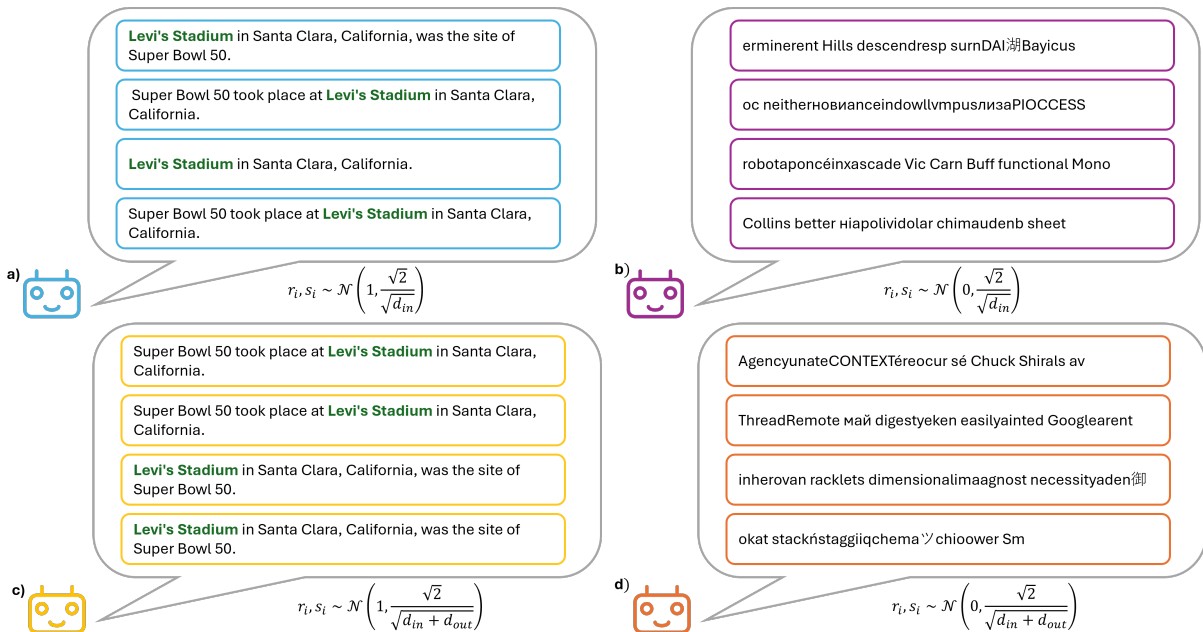

**Figure C.1.** a) He initialization with $\mu = 1$ b) He initialization c) Xavier initialization with $\mu = 1$ d) Xavier initialization

[**INST**]
# Answer the question based only on the given context. Keep the answer short. If the answer is not in the context or if you are unsure, respond with 'I don't know'.
# Context:
<The provided context >
# Question:
<A question based on the provided context >
[**/INST**]

For the MMLU dataset we instead use the following template:

[**INST**]
# Instruction
You will be given a question followed by four options: A, B, C, and D. Your response should be either A, B, C, or D.
# Question:
<A knowledge question >
# Options:
A)
B)
C)
D)
[**/INST**]

| Prompt | [INST]
Answer the question based only on the given context. Keep the answer short. If the answer is not in the context or if you are unsure, respond with 'I don't know'.
# Context
Microorganisms or toxins that successfully enter an organism encounter the cells and mechanisms of the innate immune system. The innate response is usually triggered when microbes are identified by pattern recognition receptors, which recognize components that are conserved among broad groups of microorganisms, or when damaged, injured or stressed cells send out alarm signals, many of which (but not all) are recognized by the same receptors as those that recognize pathogens. Innate immune defenses are non-specific, meaning these systems respond to pathogens in a generic way. This system does not confer long-lasting immunity against a pathogen. The innate immune system is the dominant system of host defense in most organisms.
# Question
What part of the innate immune system identifies microbes and triggers immune response?
[/INST] |
|---|---|
| Answer | pattern recognition receptors |
| Avg. En-tropy | 0.04 |

**Table D.1.** An example of a correctly extracted answer. The correct answer exists in the provided context, our model correctly identifies the answer and extracts it from the context. We note that the uncertainty is relatively low.

| Prompt | **[INST]**
Answer the question based only on the given context. Keep the answer short. If the answer is not in the context or if you are unsure, respond with 'I don't know'.
# Context
Other green spaces in the city include the Botanic Garden and the University Library garden. They have an extensive botanical collection of rare domestic and foreign plants, while a palm house in the New Orangery displays plants of subtropics from all over the world. Besides, within the city borders, there are also: Pole Mokotowskie (a big park in northern Mokotów, where was the first horse racetrack and then the airport), Park Ujazdowski (close to the Sejm and John Lennon street), Park of Culture and Rest in Powsin, by the southern city border, Park Skaryszewski by the right Vistula bank, in Praga. The oldest park in Praga, the Praga Park, was established in 1865–1871 and designed by Jan Dobrowolski. In 1927 a zoological garden (Ogród Zoologiczny) was established on the park grounds, and in 1952 a bear run, still open today.
# Question
What park is close to Vistula street?
**[/INST]** |
|---|---|
| **Answer** | Park Skaryszewski |
| **Avg. Entropy** | **0.37** |

**Table D.2.** An example of a hallucinated answer from the LLM. Specifically, note the text highlighted in orange, which states that Park Skaryszewski is located by the right *Vistula bank*. When the LLM encounters the question, "What park is close to *Vistula street*?", it incorrectly assumes that *Vistula bank* and *Vistula street* refer to the same geographical location. Consequently, it hallucinates by generating the incorrect answer of *Park Skaryszewski*, even though *Vistula street* has never been explicitly mentioned in the provided context. Consequentially, when it produces this hallucination, it produces a higher uncertainty measurement than that of the correct answer described in Table D.1.

