# OpenReview forum: "Hallucination Detection in LLMs: Fast and Memory-Efficient Fine-Tuned Models"
_NLDL.org/2025/Conference — NLDL 2025 Oral_

### Official Review · Reviewer_JRCF · 2024-10-07
**Review for hallucination detection in LLMs.**

**Confidence:** 3

**Summary:**

This paper focuses on leveraging an ensemble of Large Language Models (LLM) to generate uncertainty estimates which can be used to detect LLM hallucinations. A chief contribution was the utilisation of Low-Rank Adaptation of Large Language Models (LoRA) to train the LLM ensembles in a memory-efficient and flexible manner. The uncertainty estimates computed were used as features in binary classification models to distinguish between correct and hallucinated responses. The hallucinations were categorised into faithfulness and factual hallucinations to detect potentially harmful predictions for LLM applications in high-risk settings.

**Strengths:**

The paper's structure is good, it is well-motivated and appears to be correct in its methodology and experiments. The contributions are relevant to address existing issues regarding the faithfulness and reliability of LLMs while also making the training process more computationally efficient. The results are presented with good use of other baseline and alternative frameworks for comparison as well as good evaluation metrics. The details presented in the appendix are also beneficial to highlight specific examples of prompts and responses.

**Weaknesses:**

In the caption of the presented Figure 1, LoRA if left unabbreviated and as the figure appears before the Introduction section the unabbreviated form is not known to the user and this could be further clarified. Also, it can be further clarified in the caption that V are the fast weight matrices and the pattern behind the B and A vector boxes makes it difficult to read the text on them.

In Tables 1 and 2, highlighting the presented method such as with the BatchEnsemble (ours) would improve the clarity. Also specifying how the top-1 accuracies in Table 1 were computed from the five classfiers would be useful.

Lastly, the sentence " All models utilize..." in line 301 within Section 4 could be worded to further emphasise the model used was Mistral-7B-instruct-v.02 with pre-trained weights.

**Final Rebuttal Confidence:**

3

**Final Rebuttal Justification:**

Based on revised edition of submission I am happy to accept this submission.

**Justification:**

Overall well-motivated work with solid contributions which could be used to improve both efficiency and reliability of LLMs.

---

> ### Author Rebuttal · Authors · 2024-10-23
>
> > In the caption of the presented Figure 1, LoRA if left unabbreviated and as the figure appears before the Introduction section the unabbreviated form is not known to the user and this could be further clarified. Also, it can be further clarified in the caption that V are the fast weight matrices and the pattern behind the B and A vector boxes makes it difficult to read the text on them.
>
> Thank you for the comment. We have added an explanation of LoRA in the figure caption and clarified the variable names, including specifying that V refers to the fast weight matrices. Furthermore, we have adjusted the figure to improve the readability of the B and A matrices.
>
> > In Tables 1 and 2, highlighting the presented method such as with the BatchEnsemble (ours) would improve the clarity. Also specifying how the top-1 accuracies in Table 1 were computed from the five classfiers would be useful.
>
> Thank you for your suggestion. We have highlighted the method names using (ours) for clarity. Additionally, we have added a subsection that thoroughly explains how the five classifiers were trained. We also rephrased the first paragraph of the results section to clarify how the top-1 accuracies in Table 1 were computed.
>
> > Lastly, the sentence " All models utilize..." in line 301 within Section 4 could be worded to further emphasise the model used was Mistral-7B-instruct-v.02 with pre-trained weights.
>
> Thank you for your suggestion. We have rephrased this section to more clearly emphasize how each model utilized the Mistral-7B-Instruct-v0.2 pre-trained weights and in which way each method employed them.

---

### Official Review · Reviewer_d6y9 · 2024-10-09
**Interesting work in memory-efficient ensembling, but does it work for other models?**

**Confidence:** 4

**Summary:**

The article proposes a method to detect language model hallucinations, i.e. responses which disregard instructions or include incorrect information, based on uncertainty estimation. The authors base their approach in information theory, arguing that model uncertainty can be decomposed into epistemic uncertainty (tied to the model's knowledge of the data) and aleatoric uncertainty (representing variation in the data), and that these uncertainties are correlated with model hallucinations.

To provide uncertainty estimation for existing pretrained language models, the authors propose augmenting language models with multiple LoRA adapters, which are randomly initialized and then finetuned separately on the target task. The uncertainty is then estimated from the estimated predictive entropy of the adapters. To detect hallucinations, the authors generate a labelled dataset of uncertainty estimates and correctness labels for a set of model prompts and responses.

The authors evaluate their uncertainty estimation and hallucination detection on the Mistral-7B-Instruct-0.2 language model, and compare their approach against a baseline prompt-based method with repeated sampling, and the LoRA Ensemble method by Wang, Aitchison and Rudolph. They find that their method performs worse than LoRA Ensemble in classifying factual hallucinations and out-of-distribution examples, but performs best out of the sampled methods when classifying faithfulness hallucinations, achieving 97.8% accuracy on their test set based on SQuAD 2.0. After finetuning all of the evaluated models on questions from the SQuAD and MMLU datasets, the authors also find that their ensemble achieves better accuracy on questions from the respective test sets than the LoRA Ensemble and the original model finetuned by itself.

**Strengths:**

The proposed approach provides a straightforward and memory-efficient method to approximate a model ensemble for a pretrained language model, and to generate uncertainty estimates from it. The resulting ensemble also improves on the question answering accuracy of the original standalone model after finetuning. The approach is likely also extensible to other parameter-efficient finetuning techniques.

The authors present a reasonable hypothesis for why this approach works for faithfulness error detection. Based on the "snowball effect" where pretrained language models "commit" to continuing earlier mistakes, the authors link uncertainty for individual tokens to the model "committing" to a wrong answer.

The experimental conditions are well documented, based on publicly available benchmarks and language models, and explicitly describe the dataset processing, making the experiments easier to replicate.

**Weaknesses:**

The approach is presented as generally applicable to instruction-tuned language models with decoders - however, the experiments only evaluate the approaches on a single pretrained language model. Evaluations with multiple pretrained language models are necessary to establish how the hallucination detection performs with weaker or stronger pretrained models, and could also provide confidence estimates for the reported results.

While the experimental conditions themselves are well documented, the parameters for the LoRA ensemble itself - such as the number of adapters and their size - are not given in the article. Since these parameters present a tradeoff between inference time, memory use and the results on the downstream tasks, omitting them from the article makes the results challenging to replicate. Additionally, it is not clear which dataset the training set for the hallucination classifiers is derived from.

Finally, I think the paper would be stronger by referencing model editing techniques such as ROME [1], which attempt to change factual and conceptual associations in language models, while retaining their overall question answering capabilities. Ideally, these techniques could generate models with more factual errors but the same instruction following capabilities as the original, allowing experiments to substantiate the authors' hypothesis that faithfulness and factual errors are correlated with the estimated aleatoric and epistemic uncertainty.

Additional questions which did not significantly impact the decision:

* In the "Uncertainty Estimation" subsection, does $\mathcal{D}$ refer to the overall data distribution?

* Since the weight decay of the LoRA Ensemble and noise injection lead to worse predictive results, are there any objectives or mechanisms in the main method which maintain diversity among the ensemble members (avoiding the case where B and A in all LoRA ensemble members go to zero?)

* Since the faithfulness error detection is based on unanswerable questions from SQuAD 2.0, including factual questions with definite answers outside the question context, are there faithfulness hallucinations where the model is factually correct? Do the uncertainty estimates reflect this?


[1] Meng, K., Bau, D., Andonian, A., & Belinkov, Y. (2022). Locating and editing factual associations in GPT. Advances in Neural Information Processing Systems, 35, 17359-17372.

**Final Rebuttal Confidence:**

4

**Final Rebuttal Justification:**

The authors' revisions include necessary details about the ensemble setup during their experiments, which support their argument that the LoRA adapters are serving as a memory-efficient approximation of a full model ensemble. Additionally, the authors have clarified key points, and thoroughly addressed the reviewers' questions and suggestions. While I would like a larger evaluation of the approach with different ensemble hyperparameters and base language models, the authors acknowledge this as a direction for future work, and their detailed experimental descriptions would also allow other researchers to perform this evaluation.

With these revisions, I am comfortable accepting this paper, and would like to see it presented at the conference.

**Justification:**

The paper is overall well written and easy to follow, and presents an easily adaptable and memory-efficient method to provide joint uncertainty and faithfulness error detection for existing pretrained language models. I think this paper could be a very good reference point for further work in the overlap between parameter-efficient finetuning and downstream use of uncertainty estimation.

However, the paper omits key details about the ensemble design and training of the LoRA adapters, making it challenging to recreate the results. This also makes it difficult to exactly quantify the benefit to memory efficiency, one of the main stated benefits of the method.

Since the method is proposed as a general-purpose uncertainty estimation and hallucination detection method, I also find it a significant weakness to only evaluate the approach on one pretrained model, when evaluations on other models would both bolster confidence in the method, and provide important evidence to corroborate the authors' hypothesis linking errors to aleatoric and epistemic uncertainty.

Unfortunately, while I found the paper interesting and the ideas worth exploring further, I am not comfortable accepting it as-is. I would however be comfortable revising my rating to an accept if the ensemble details are included in the revised article.

---

> ### Author Rebuttal · Authors · 2024-10-23
>
> > The approach is presented as generally applicable to instruction-tuned language models with decoders - however, the experiments only evaluate the approaches on a single pretrained language model. Evaluations with multiple pretrained language models are necessary to establish how the hallucination detection performs with weaker or stronger pretrained models, and could also provide confidence estimates for the reported results.
>
> Thank you for your insightful feedback. We agree that evaluating our approach across multiple pretrained language models with varying parameter sizes—representing weaker or stronger models—would provide valuable insights into how the uncertainty-based method performs under different knowledge capacities. While our method is indeed applicable to architectures beyond Mistral 7B, implementing it across different models would require adapting the codebase for each specific architecture. Additionally, due to the constraints of our computational budget, conducting such extensive evaluations is beyond our current resources. We have added a paragraph in the discussion of this, as we recognize the importance and consider it a valuable direction for future work.
>
> > While the experimental conditions themselves are well documented, the parameters for the LoRA ensemble itself - such as the number of adapters and their size - are not given in the article. Since these parameters present a tradeoff between inference time, memory use and the results on the downstream tasks, omitting them from the article makes the results challenging to replicate. Additionally, it is not clear which dataset the training set for the hallucination classifiers is derived from.
>
> Thank you for your valuable comments. We apologize for omitting the specific parameters used in the LoRA ensemble, and have added this in the revised version. To clarify, for the LoRA ensemble, we used a lora rank 8 and a scaling factor α=32, with each LoRA member containing 54,525,952 trainable parameters. Our ensemble size was set to 4, resulting in a total of approximately 218 million trainable parameters for the entire LoRA ensemble. For the Batch Ensemble, we solely use the LoRA matrices during training, after which we merge the LoRA matrices into the shared weight. Hence, during inference, only the fast weight parameters are applied, which are smaller since they consist of rank 1 matrices.
>
> Regarding the lack of clarity on the training details of the hallucination classifiers, we acknowledge this oversight and have addressed it in the revised version with a dedicated subsection. This section outlines how the five classifiers were trained and how the datasets were created. Briefly, for all three tasks, 5000 model predictions were annotated as hallucinated or non-hallucinated based on task-specific criteria, with an 80/20 training/testing split, using a random seed of 42.
>
> > Finally, I think the paper would be stronger by referencing model editing techniques such as ROME [1], which attempt to change factual and conceptual associations in language models, while retaining their overall question answering capabilities. Ideally, these techniques could generate models with more factual errors but the same instruction following capabilities as the original, allowing experiments to substantiate the authors' hypothesis that faithfulness and factual errors are correlated with the estimated aleatoric and epistemic uncertainty.
>
> Thank you for your insightful suggestion regarding model editing techniques such as ROME. While ROME is not directly related to hallucination detection, we agree that your point raises an interesting avenue for further exploration. Initially, we hypothesized that factual hallucinations might be more closely linked to epistemic uncertainty than aleatoric uncertainty. Our goal was to design a Bayesian logistic regression model that could classify not only whether a hallucination occurred or not, but also distinguish between different types of hallucinations—factual or faithful. During this process, we considered how to handle cases where a model might simultaneously exhibit both types of hallucinations. The approach you suggest—using model editing techniques like ROME to retain instruction-following capabilities while slightly increasing factual errors—would indeed provide a valuable way to evaluate the correlation between epistemic uncertainty and factual hallucinations. This could be a promising area for future research to investigate if our initial hypothesis holds.
>
> > In the "Uncertainty Estimation" subsection, does D refer to the overall data distribution?
>
> That is correct, D corresponds to the training data distribution. In the revised version, we have clarified this when describing Equation (1).
> > Since the weight decay of the LoRA Ensemble and noise injection lead to worse predictive results, are there any objectives or mechanisms in the main method which maintain diversity among the ensemble members (avoiding the case where B and A in all LoRA ensemble members go to zero?)
>
> Thank you for your question. In the main method, we have not explicitly incorporated any objective or regularization term to maintain diversity among the ensemble members. Instead, the observed diversity appears to arise from the stochasticity introduced during the initialization of the LoRA A matrix and the fast weights for each ensemble member. While we acknowledge that explicitly enforcing diversity could be beneficial, our attempts to do so using noise injection did not yield satisfactory results. Therefore, identifying a suitable approach to promote diversity without compromising performance remains an open question for future research.
> > Since the faithfulness error detection is based on unanswerable questions from SQuAD 2.0, including factual questions with definite answers outside the question context, are there faithfulness hallucinations where the model is factually correct? Do the uncertainty estimates reflect this?
>
> Thank you for your thoughtful question. Indeed, we observed instances where the model produces faithful hallucinations that are factually correct. This observation was one of the main motivations behind our investigation into whether epistemic or aleatoric uncertainty correlates with factual and faithfulness hallucinations. Although our initial exploration did not yield conclusive results, we believe that further research, particularly in developing more diverse ensembles, may help uncover this relationship. Regarding your second question, we did not systematically verify the factual correctness of faithfully hallucinated responses, as doing so would require an extensive manual annotation process against external sources.

---

### Official Review · Reviewer_TshU · 2024-10-10
**Hallucination deteciton in LLMs**

**Confidence:** 4

**Summary:**

The method proposed uses entropy to estimate uncertainty in LLM outputs and use this as input in various ensemblems to predict whetehrr an LLM is halluciating or not.

**Strengths:**

The paper demonstrates usefulness of uncertainty estimates (to some extend) and how best to combine these for hallucination classification in LLMs. There is no evidence in the paper that the obtained results compare to other hallucination detection strategies.

**Weaknesses:**

The method variations are not compared to other hallucination detection methods (referenced in the paper), only to different strategies for combinning and useing the unceretainty metrics. I recoommend as a minimum to compare to results obtained in those other papers in the discussion/conclusion.

**Justification:**

There is a lack of comparison to existing literature on hallucination detection.

---

> ### Author Rebuttal · Authors · 2024-10-23
>
> > The method variations are not compared to other hallucination detection methods (referenced in the paper), only to different strategies for combinning and useing the unceretainty metrics. I recoommend as a minimum to compare to results obtained in those other papers in the discussion/conclusion.
>
> Thank you for your feedback. Our approach focuses on uncertainty-based methods, as these are most similar to the method we propose. Specifically, uncertainty-based methods are versatile and can be applied to both factual and faithfulness hallucinations, making them directly comparable to our approach in both categories.
>
> The other methods referenced in the paper are designed specifically to detect either faithfulness hallucinations (e.g., Kryscinski et al. [3] and Laban et al. [5]) or factual hallucinations (e.g., Huo et al. [19] and Min et al. [21]), where adherence to instructions and contextual cues or the verification of outputs from reliable sources is paramount. These methods are task-specific and excel at addressing one particular type of hallucination but are not as versatile when faced with other types. For instance, it is not trivial to solve factual hallucination occurrences with a method specifically designed to tackle faithfulness hallucinations, and vice-versa.
>
> While we acknowledge that task-specific methods might outperform uncertainty-based methods in their particular areas of focus, our aim was to develop and evaluate a more generalizable tool. Therefore, we chose to focus on comparing our approach to other uncertainty-based methods, as they offer broader applicability across different types of hallucinations.
>
> We hope this clarifies our intention in selecting comparison baselines, and we have updated the discussion of our manuscript to better inform the readers who would express similar concerns as yours.
>
> [3] Kryściński, W., McCann, B., Xiong, C. and Socher, R., 2019. Evaluating the factual consistency of abstractive text summarization.
> [5] Laban, P., Schnabel, T., Bennett, P.N. and Hearst, M.A., 2022. SummaC: Re-visiting NLI-based models for inconsistency detection in summarization.
> [19] Huo, S., Arabzadeh, N. and Clarke, C.L., 2023. Retrieving supporting evidence for llms generated answers.
> [21] Min, S., Krishna, K., Lyu, X., Lewis, M., Yih, W.T., Koh, P.W., Iyyer, M., Zettlemoyer, L. and Hajishirzi, H., 2023. Factscore: Fine-grained atomic evaluation of factual precision in long form text generation.

---

### Meta-Review · Area_Chair_CKcc · 2024-11-02

**Recommendation:** Accept (Oral)
**Confidence:** 4

**Metareview:**

The paper aims to detect hallucinations of LLM by estimating the uncertainty of the model.

The reviewers raised several questions which were mostly answered by the authors, most of which were about clarification of the text.

I recommend accepting the paper for a presentation.

**Suggested Changes To The Recommendation:**

1: I agree that the recommendation could be moved down

---

### Decision · Program_Chairs · 2024-11-06

**Decision:**

Accept (Oral)

**Comment:**

We recommend an oral and a poster presentation given the AC and reviewers recommendations.